# Management Strategies for Antipsychotic-Related Sexual Dysfunction: A Clinical Approach

**DOI:** 10.3390/jcm10020308

**Published:** 2021-01-15

**Authors:** Angel L. Montejo, Rubén de Alarcón, Nieves Prieto, José Mª Acosta, Bárbara Buch, Laura Montejo

**Affiliations:** 1Psychiatry Service, Clinical Hospital of the University of Salamanca, 37007 Salamanca, Spain; nievesprieto2010@gmail.com; 2Institute of Biomedical Research of Salamanca (IBSAL), Paseo San Vicente SN, 37007 Salamanca, Spain; jmacosta@saludcastillayleon.es (J.M.A.); bbuch@saludcastillayleon.es (B.B.); 3Nursing School, University of Salamanca, Av. Donates de Sangre SN, 37007 Salamanca, Spain; 4Psychiatry Service, Infanta Elena Hospital, Juan Ramon Jiménez Hospital, Ronda Exterior Norte S/N, 21080 Huelva, Spain; ruperghost@gmail.com; 5Barcelona Bipolar and Depressive Disorders Program, Institute of Neurosciences, University of Barcelona, IDIBAPS, CIBERSAM, Hospital Clinic of Barcelona, 08401 Catalonia, Spain; laumonteg@gmail.com

**Keywords:** antipsychotic, sexual dysfunction, erectile dysfunction, anorgasmia, orgasm retardation, TESD, management strategies, treatment

## Abstract

Antipsychotic medication can be often associated with sexual dysfunction (SD). Given its intimate nature, treatment emergent sexual dysfunction (TESD) remains underestimated in clinical practice. However, psychotic patients consider sexual issues as important as first rank psychotic symptoms, and their disenchantment with TESD can lead to important patient distress and treatment drop-out. In this paper, we detail some management strategies for TESD from a clinical perspective, ranging from prevention (carefully choosing an antipsychotic with a low rate of TESD) to possible pharmacological interventions aimed at improving patients’ tolerability when TESD is present. The suggested recommendations include the following: prescribing either aripiprazole or another dopaminergic agonist as a first option antipsychotic or switching to it whenever possible. Whenever this is not possible, adjunctive treatment with aripiprazole seems to also be beneficial for reducing TESD. Some antipsychotics, like olanzapine, quetiapine, or ziprasidone, have less impact on sexual function than others, so they are an optimal second choice. Finally, a variety of useful strategies (such as the addition of sildenafil) are also described where the previous ones cannot be applied, although they may not yield as optimal results.

## 1. Introduction

Sexual dysfunction (SD) as a consequence of treatment with psychotropic drugs, especially antipsychotics (APS), is a common adverse effect [1,2,3,4,5,6]. Its acknowledgement, as well as its clinical management, has become a goal of utmost importance for the clinician, as it often jeopardizes long-term adherence [7,8,9,10,11,12,13,14].

Ever since the first reports of chlorpromazine and thioridazine causing orgasmic and erectile dysfunction were published [15,16], the knowledge of treatment emergent sexual dysfunction (TESD) and its underlying mechanisms of action has increased greatly. Human sexuality is affected by a variety of neurogenic, psychogenic, vascular, and hormonal factors [17], and the action of antipsychotics may unbalance some of these mechanisms [18]. For example, dopamine facilitates sexual function through the mesolimbic system [19,20], so strict anti-dopaminergic antipsychotics can obstruct this process, but partial dopaminergic agonists such as aripiprazole and cariprazine may not. On the other hand, serotonin inhibits sexual desire [21,22,23,24], likely through postsynaptic 5HT2 receptors, which are often targeted by atypical antipsychotics [18,25] and can disrupt a complex balance between stimulation and blockade that affects sexual arousal, orgasm, and ejaculation retardation. Noradrenalin increases the ability for arousal through its influence on central receptors and inhibits erection by binding with peripheral α1 receptors [18,26]. α1-adrenergic receptors contribute to the emptying of cavernous bodies, so their stimulation can cause erectile dysfunction and their blockade can cause priapism [27,28]. Their counter-regulatory mechanisms involve the activation of cholinergic fibers, so anticholinergic drugs (such as some APS at certain doses [29]) would cause erectile dysfunction [30,31].

Moreover, neuroendocrine changes [32] involving the luteinizing hormone (LH) [33] and the follicle-stimulating hormone (FSH), as well as long-term high prolactin blood levels [34,35,36] and low testosterone [37,38], also play a part on antipsychotics, causing SD, whether directly or indirectly [39]. Finally, neural nitric oxide (NO) has been known to also regulate sexual behavior and erectile function, and its expression in the hypothalamus could be potentially blocked by some antipsychotics [40,41].

SD may cause diverse clinical alterations, such as low libido, difficulties in ejaculation, difficulties reaching orgasm, erection and vaginal lubrication, as well as menstrual alterations or gynecomastia [42,43]. These alterations are mostly reversible with treatment discontinuation, except for priapism, which can require surgical intervention in some cases [18,44].

Most psychiatrists frequently underestimate the presence of sexual dysfunction, even when adjusting for different SD prevalence across populations [45]. It is not always considered a priority from a clinical perspective, and it is often disregarded almost completely [1,46,47]. However, when patients are asked directly about SD, they rank it as a relevant problem. In a study by Finn [48], where the costs of drug side effects and symptoms among psychotic patients were measured, they regarded the presence of SD (inhibited/painful ejaculation and impotence) as the most important drug side effect (over common ones like weight gain, oversedation, or hypotension), and just as important as suffering from persecutory delusions and other positive symptoms.

We must be mindful that SD usually develops in complex ways that involve several factors (comorbidity, culture, addictions, medication, and so on) [42], and is the consequence of the interplay of these factors with one another. For example, a recent study suggests that the severity of psychosis alone might contribute to SD independently of APS [49]. The diversity of these factors can be often ignored or dismissed on SD clinical studies. However, most of the current clinical literature focuses on SD in psychotic patients because of suboptimal drug treatment [50], researching strategies to diminish or remove TESD from this perspective.

Depending on whether or not specific interviews are used for detecting SD, incidence rates of TESD can vary greatly. Information printed on technical data sheets for each APS is usually unquantified, or grossly underestimates TESD frequency [51]. These data are gathered from either clinical trials or post-commercialization observations, which mainly aim to measure clinical efficacy, not SD incidence. However, when specific interviews are used to measure SD, the incidence rates are consistently higher, at 40–50% [52,53]. Old samples of chronic schizophrenic patients in outpatient monitoring suggest up to 60% of SD in this population after years of treatment [54].

The abysmal difference between these rates can be attributed to patients poorly communicating this issue, and the fact that they do not report each case of TESD they experience. Multiple explanations are behind this, including shame, poor patient–clinician rapport, or physicians who have difficulties in discussing sexuality with their patients [42,55]. All these reasons contribute to the incidence of SD being underestimated and underreported [46,56,57]. This issue going unnoticed can cause difficulties in interpersonal relationships and sexual dissatisfaction for patients and their partners. This could explain many cases of low adherence wrongly interpreted as a “tolerance to treatment”. To incorporate this issue into routine clinical practice would be of great benefit to patients [58,59], as has been recommended in a recent multidisciplinary consensus [60].

When analyzing SD incidence rates by antipsychotic in a recent meta-analysis (Table 1), it becomes clear that their different mechanisms of action may contribute to separate profiles of adverse effects and how they affect each sexual dimension [61,62,63,64]. However, prevalence of SD in Table 1 does not directly correlate to the incidence rates of TESD by each antipsychotic, because of the influence of several confounding factors (i.e., concomitant medication) [65].

Given the wide array of APS available and their potential for TESD, we aimed in this paper to gather and update relevant information about possible management strategies for this issue that can be considered effective enough without compromising the antipsychotic effect.

## 2. Methodology

We performed a review of TESD with antipsychotics and its possible management strategies. PRISMA guidelines were followed when possible (Figure 1). There was no specific time-delimitation, but priority was given to literature reviews and articles that had been published most recently, with special attention given to articles that had been subjected to peer review. PubMed and Cochrane were the databases used. The term “antipsychotic” was used along with Boolean operator “AND” in combination with several terms: “sexual dysfunction”, “erectile dysfunction”, “sexual function*” (to allow for both “function” and “functioning”), “desire”, “orgasm”, and “anorgasmia”.

We excluded articles that focused on SD as a general entity rather than as a consequence of treatment with APS, as well as those articles with a broader scope on antipsychotics various side effects that did not focus on sexual function. We also excluded most articles that heavily focused only on metabolic alterations (primarily, hyperprolactinemia or lack thereof) because of antipsychotic treatment, instead of the clinical relevance of TESD. We considered primarily articles written in English, Spanish, Portuguese, or French, although at least one key article written in Dutch was considered for inclusion.

Initially, 1242 relevant results were found. After applying exclusion criteria via abstract screening, 186 relevant results remained; after full manual screening, they were reduced to 178, finding 15 more relevant results through cross-referencing. Upon manual screening and categorizing, these results included (1) those that acknowledged the prevalence and clinical relevance of the phenomenon of antipsychotic TESD, and (2) those that focused on clinical management strategies for TESD.

The results described under “(1)” were mainly used for Section 1 of this paper. The results on “(1)” added up to a total of 45 articles, including 15 transversal and cross-sectional studies, 11 observational studies, and a total of 19 reviews (of which 1 of them was a meta-analysis). Several of these articles were also used on other sections of the paper to provide background, as some of them are antipsychotic-specific.

The results described under “(2)” are included in Section 3 of this paper. There were 121 articles, including 25 reviews (of which 4 were systematic reviews or meta-analyses), 21 experimental studies, 44 observational studies, and 31 papers featuring case series or case reports. The Zotero reference management tool was used to build a database of all the considered articles.

## 3. Results

When dealing with TESD, it is practical to be mindful of a three-step sequence: (1) Preventing TESD in the susceptible population. (2) Conducting routine checks for TESD in sexually active patients who are prescribed APS. (3) Performing clinical interventions when TESD is a problem for the patient or poses a risk for treatment drop-out.

### 3.1. Primary Prevention: Using APS with a Low Incidence of TESD

While many patients may not be on monotherapy and SD may emerge from a combination of drugs, prevention allows us to optimize the risk for SD when treating an antipsychotic-naïve patient. Although keeping first rank psychotic symptoms under control should be a priority, clinicians should always be wary of the potential for TESD of each drug at this point (Table 2). Overall, risperidone [66,67] and other classical APS [62,68,69,70,71,72,73] have a higher incidence of TESD when compared with other atypical neuroleptics such as clozapine [73,74,75], olanzapine [62,68,71,76,77,78,79,80,81,82], quetiapine [62,66,71,72,83,84,85,86], or aripiprazole [87,88,89]. When analyzing TESD by each sexual dimension comparing different APS, this becomes even more obvious, given how SD rates are so different for each antipsychotic [52].

Risperidone has been associated with other problems in addition to regular sexual dysfunction, like retrograde ejaculation and urinary dysfunctions [90,91,92,93,94]; anejaculation [95,96]; and, on rare occasions, hypersexuality [97,98]. Part of this SD could be related to high levels of blood prolactin [84,99], resulting in complex hormone regulations that decrease testosterone [100]. Some studies question prolactin levels as the sole mechanism of SD in risperidone [34,65,83,101], and it is likely that a blockade of 5HT2A and 5HT2C receptors at the prefrontal cortex may be responsible too [34,102,103]. Nonetheless, hyperprolactinemia also causes a variety of adverse effects and pathologies that go beyond sexual dysfunction, especially in the long-term [1,60].

Paliperidone, a metabolite of risperidone, has also been linked to hyperprolactinemia [104] and sexual dysfunction [105,106], with similar mechanisms involved. Therefore, hyperprolactinemic APS such as risperidone, paliperidone, amisulpride, and classical antipsychotics are not an appropriate first option if we wish to keep sexual function preserved in psychotic patients [1,65]. In turn, the other atypical antipsychotics fare slightly better in comparison [70].

Clozapine’s weaker blockade of D2 dopaminergic receptors in the mesostriatum is supposed to have a lesser incidence of orgasm dysfunction and erectile dysfunction when compared with typical neuroleptics [107,108], although not by a wide margin [109]. A case of hypersexual behavior with clozapine has been reported [110], although it is not a common adverse effect.

Ziprasidone, because of its mechanism of action (5-HT2 receptor antagonism facilitating dopamine release in the cortex), would theoretically improve orgasmic and erectile function, as suggested by some case reports concerning spontaneous ejaculation and priapism [111,112]. However, there is only one study researching its effect on sexual function to our knowledge. In a small (*n* = 15), 3-month observational study, patients who were antipsychotic-naïve commenced treatment with ziprasidone, preserving sexual functioning across all sexual dimensions, and even slightly improving the minor sexual dysfunction found in baseline [113]. Although promising, these results need further confirmation.

Olanzapine causes less extrapyramidal symptoms [76], less hyperprolactinemia [79,80,114], and less SD than haloperidol [115] and risperidone [68,71,81,116]. Orgasm, ejaculation, and arousal are usually less affected by olanzapine than by risperidone [78,81], as also suggested by some case reports of spontaneous ejaculation [117] and priapism [118]. However, some studies consider these differences to be relatively minor or nonexistent [66,114], especially when olanzapine doses reach 15–20 mg/day [62,82]. These results suggest that olanzapine causes a dose-dependent mild to moderate hyperprolactinemia.

Quetiapine, on the other hand, does not significantly increase prolactin levels [82]. There has been lower reported incidence of SD with quetiapine when compared with typical neuroleptics, and some atypical (risperidone and olanzapine) [72,83,84,119]. With one exception [72], most of the studies agree that, when dosing below 400 mg/day, sexual function is mostly preserved [62,82,84], but when dosing at around 500 mg/day and above, SD may appear with a severity in proportion to dosage [66,120], although usually less so than the previously mentioned antipsychotics [53,121]. Quetiapine has also been described as particularly helpful in preserving sexual arousal [52,66,72].

Aripiprazole, a partial dopamine receptors agonist, as well as its partial 5HT1A agonism and 5HT2A antagonism, has been correlated to no actual increase in prolactin levels [88,119,122]. In fact, when used in combination with another antipsychotic, it has been observed to lower already increased prolactin levels [123]. When dosing at around 15 mg/day, this results in no SD at all [53,87,88], or SD of mild intensity, usually being tolerable for the patient. One study concludes that aripiprazole causes less SD than risperidone, olanzapine, or quetiapine [119]. However, while aripiprazole remains the best option among the researched evidence when trying to prevent TESD [70], several cases of hypersexual behavior have been reported to be directly caused by this drug [123,124,125,126,127], which apparently can manifest at any dose, but seem to be more frequent when dosing above 20 mg/day [125]. Cariprazine, which is another dopamine partial agonist like aripiprazole that does not produce hyperprolactinemia [128,129], still has no published data regarding its effect on sexual function. Although it could be considered similar to aripiprazole, specific studies are needed.

### 3.2. Detection and Exploration of TESD

Psychotic patients are a very heterogeneous group in terms of clinical evolutions, treatment response, and sexual activity, so being mindful of this issue and its possible repercussions on adherence becomes difficult at times for both the clinician [46] and the patient. For example, uncompensated patients with acute symptomatology that require hospitalization do not share the same concern for their sexuality as fully stabilized chronic patients do [130,131,132]; clinicians treating the former may overlook this issue at first while attempting to treat psychotic symptomatology.

That is why, when SD shows up, it is crucial to examine how the patient accepts this adverse effect and how it influences their quality of life and overall functionality. The spontaneous communication rate of SD from patients is low [1,53,133], so medical staff should routinely check with the patients to detect if TESD exists by asking clear questions, such as “How has your sexual life been since you started taking the medication?” and “Have you noticed any change that worries you?” This issue should always be approached in a neutral manner, expressing concern about the patient’s wellbeing and informing them about the potential sexual side effects of APS medication [59]. Gathering a full sexual medical history is useful to thoroughly understand the onset of the first SD symptoms, SD severity, the impact of SD on the sexual lives of the patient and their partner, and the influence of SD on the patient’s quality of life [134]. This thorough evaluation would also help to determine whether SD is in fact a consequence of antipsychotic treatment or is caused by other factors such as comorbidities, substance abuse, or other prescribed medication [3].

To better quantify TESD and measure its degree of acceptance by the patient, several specific validated scales and structured questionnaires can be of use. As the rate of SD can vary depending on which scale is used [68], a combination of several of them would be best in order to best determine its existence and its intensity, especially when researching this issue [65].

A few suitable options for measuring TESD caused by antipsychotics are the following [135]: the Psychotropic-Related Sexual Dysfunction Questionnaire (PRSexDQ-SALSEX, free download at http://sexualidadysaludmental.com/salsex.html)) [136,137]; the Changes in Sexual Functioning Questionnaire (CSFQ) [138]; the Antipsychotics and Sexual Functioning Questionnaire (ASFQ) [139]; the Arizona Sexual Experiences Scale (ASEX) [140,141,142]; the Nagoya Sexual Function Questionnaire (NSFQ) [143]; and the Antipsychotic Non-Neurological Side Effects Rating Scale for sexual dysfunction (ANNSERS) [144]. The PRSexDQ, ASFQ, and CSFQ cover all stages of sexual functioning, which makes them preferrable [135]. To further measure the severity of SD, the International Index of Erectile Function (IIEF) and the Sexual Encounter Profile (SEP) [145] are valid options for men, and the Sexual Interest and Desire Inventory-Female (SIDI-F) for women [146].

### 3.3. Intervention

Once TESD is identified, if it is poorly tolerated by the patient and risks compromising adherence, it may be possible to intervene depending on the particular clinical symptoms of each case. While sometimes, fully restoring previous sexual function is not achievable, partially alleviating the experienced symptoms can satisfy the patient’s expectations without jeopardizing the antipsychotic effect of their treatment. When compared with SD caused by antidepressants, the literature on intervention in TESD caused by APS is scarce, and frequently inconsistent [2,147], although some viable options emerge. Some possible strategies for treating TESD are the following: (1) waiting for spontaneous remission; (2) dose reduction; (3) switching to another APS; and (4) the addition of an antidote (coadjuvant drug). None of these methods is devoid of risks, such as a relapse, so a careful individualized approach is suggested so that patients can make an informed choice.

#### 3.3.1. Waiting for Spontaneous SD Remission

Sometimes, adverse effects subside over time. Spontaneous SD remission has been reported in patients with antidepressant treatment [148]. However, it only has been reported to occur in 5–10% of the subjects and generally after 4 to 6 months of treatment [2]. One study analyzed this possibility with antipsychotics [108], and found that patients treated with clozapine experienced TESD, but it decreased dramatically or completely resolved over the course of 18 weeks. In our Spanish Working Group for the Study of Psychotropic-Related Sexual Dysfunction, this adaptation or tolerance to the adverse effect was also analyzed [149]. We found 6 out of 36 patients (16.7%) with SD caused by risperidone presenting spontaneous improvement after 6 months, and one patient improving after 3 months. These results prove unlikely that spontaneous remission can be expected before this time.

An alternative explanation for SD spontaneously improving over time would be that SD is more directly related to the severity of illness rather than the antipsychotic treatment; thus, after being treated, sexual function would consequently improve.

However, considering the current data, this strategy does not seem to be recommendable. It possibly encourages treatment drop-out, thus deteriorating quality of life and disrupting the patient–physician relationship [2,150].

#### 3.3.2. Dose Reduction

As seen with selective serotonin reuptake inhibitors (SSRIs) [2], dose reduction strategies could work in theory [151,152], as the diverse effects of APS are highly dependent on dosage. An alternative explanation could be that illness severity is related to SD, and as the decision of a lower dosage implies a clinical improvement, psychosis may not impact sexual function as severely as previous higher doses of antipsychotic.

Nevertheless, some authors recommend decreasing the dose as the first step in intervention [153], but this is not always effective [154] and may in fact risk a relapse [1]. Therefore, perhaps only selected patients with relatively mild symptoms or an overall good prognosis may be candidates for this strategy. In a study by our Spanish Working Group, we found that 19 out of 26 patients (73%) treated with risperidone, who presented SD, improved partially or fully when the dose was reduced to 50% [149]. Individual differences play a key part here, and when comparing multiple studies mentioned in Section 3.1, a certain dose of one specific antipsychotic does not always equate to a proportional correlation of SD in all patients. This might suggest that, above a certain threshold dose, patients with APS may develop SD of varying degrees independently of dose increase. This can be seen when comparing the rates of SD caused by olanzapine and risperidone at different doses (Table 3).

A strategy similar to dose reduction that has sometimes worked when dealing with TESD caused by antidepressants is the periodic treatment interruption or “weekend holiday” [148], where the patient interrupts treatment for a period of time (24 h prior to intercourse). The usefulness of this strategy is highly dependent on the specific half-life of the drug being used. No studies have been performed with APS regarding this strategy [2,156]. In fact, randomized controlled trials are currently needed to provide evidence for this or any other dose reduction strategies [147,157].

While perhaps useful regarding antidepressant treatment, this strategy is not recommended in Schizophrenia spectrum disorders as it carries significant risks of relapse, and may inadvertently encourage treatment drop-out.

#### 3.3.3. Switching to Another Antipsychotic

In patients who present severe SD who cannot risk reducing treatment doses, or are unwilling to continue treatment because of this cause, switching to another antipsychotic with a lesser SD incidence rate is a viable option [3,153].

Most authors agree on aripiprazole as the best option in this case [1,89,119,134,150,157], as several studies have proven that it improves SD rates across all sexual dimensions (desire, arousal, and orgasm) and decreases prolactin when switching from previous antipsychotics such as typical ones [158,159], risperidone [88,154,158,159,160,161], paliperidone [106], amisulpride [158,159,160], quetiapine [159], olanzapine [158,159], and clozapine [159]. The only study assessing the issue of TESD in long-acting intramuscular presentations [106] corroborates this recommendation when comparing aripiprazole to paliperidone palmitate.

There are other switching options to decrease SD, although the evidence for them is not as strong, as most studies have focused on aripiprazole’s potential.

A small open label trial points to olanzapine as an option [147], as it had been previously suggested by one case report [129] and an observational study [155].

As for quetiapine, the results are contradictory. Case reports and prospective studies suggest that switching to quetiapine might improve SD in a way that is tolerated by the patient [162,163], with one randomized trial confirming these positive results [84] at a mean dose below 500 mg/day. However, another randomized trial did not show a significant difference in SD improvement [63,147]. Nonetheless, it is still recommended as an option by some authors [1,150,157].

Further less-researched options include ziprasidone, with a 3-month prospective switching study showing improvement in SD [113]. Lurasidone, with a case report showing a decrease in SD when switching from risperidone [90] and a clinical trial showing it was not associated with treatment-related SD versus placebo [164], could be a useful alternative. Nonetheless, hypersexuality has been associated with both lurasidone [165] and ziprasidone [166], so additional research is needed. Unfortunately, a recent clinical trial aimed at knowing the outcome of the antipsychotic switching strategy versus maintaining prior treatment failed to obtain significant results because of a very low level of recruitment [167], probably explained by a reluctance among patients to switch medication and reticence of both staff and patients to talk about sex. Now, there are no studies of switching to cariprazine, which has a mechanism of action similar to aripiprazole, although theoretically, it could be an appropriate alternative [128,168]. Finally, brexpiprazole, a metabolite of aripiprazole, has been shown in clinical trials not to modify prolactin levels [169], and possibly has a good profile on sexual function.

#### 3.3.4. Use of Antidotes or Coadjuvant Treatment

When switching to another antipsychotic is not recommended or there is a potential risk for relapse, the next strategy could be the prescription of another drug or “antidote” to improve TESD in some way. In addition to being the best switching option when dealing with TESD, aripiprazole has been noted to both decrease prolactin levels and improve SD when taken concomitantly with other antipsychotics [1,2,170,171]. This has been observed across populations by several randomized studies [159,172,173] when acting as coadjuvant treatment for a variety of APS with different mechanisms of action, both typical and atypical. Aripiprazole’s action as a partial dopaminergic agonist and its role in reversing hyperprolactinemia are probably responsible for improving TESD [2], perhaps through the increase of dopaminergic activity. Although the specific dosage may be individualized according to TESD severity and the other APS prescribed, aripiprazole has been well tolerated as an add-on drug without worsening psychiatric symptoms in any way [159,172].

Aside from aripiprazole, phospodiesterase-5 (PDE-5) inhibitors have been observed to improve erectile dysfunction (ED) caused by APS [1,2,3,134,174,175]. Their mechanism of action targets the nitric oxide inhibition that some APS cause [40,41]. After the first successful case report with sildenafil, suggesting it as an option [176], two small observational studies followed with moderate to good results [177,178]. Despite a small sample (*n* = 32) and short duration (2 weeks), a randomized placebo-controlled trial confirmed the potential of sildenafil as coadjuvant treatment in TESD [179], as ratified by a Cochrane review [147]. As with aripiprazole, the specific dose probably needs to be individualized for each patient, with some experiencing improvement at 25 mg/day and others at 100 mg/day. Furthermore, a successful case report with tadalafil [180] also inspired another randomized placebo-controlled trial with good results using the ASFQ [181], despite a similarly small sample (*n* = 15) owing to it being a crossover pilot study. Other PDE-5 inhibitors have also been studied for this purpose, such as vardenafil, with good results [182], as well as lodenafil carbonate [183], which did not render statistical differences when compared with placebo.

Regarding other add-on options, the evidence is not as strong. The coadjuvant use of pro-dopaminergic drugs has also theoretically been proposed as a solution [2,3,184,185], but given the risk of worsening psychiatric symptoms, they are not recommended in this case. The only study we found with these types of drugs for TESD was a small placebo-controlled study with selegiline, which did not seem to improve SD, despite decreasing prolactin levels [186].

Other small observational studies and cases series also suggest improvement of TESD with the addition of imipramine at lower doses (25–50 mg/day) [187,188], mirtazapine (30 mg/day) [189], saikokaryukotsuboreito (Herbal Medicine) [190], and pramipexol (0.25 mg) [191].

## 4. Clinical Recommendations

Although there is scarce information, it seems that the incidence of antipsychotic related TESD is deeply underestimated by clinicians. The main reason is that psychiatrists rarely question their patients about their sexual activity, considering it erroneously intrusive; a lack of interest, time, or training in these competencies is also to blame. Antipsychotic treatment is usually necessary for long periods of time or even indefinitely, and SD can compromise adherence [11,12,192], thus worsening the patient’s quality of life [106] as well as their partner’s, leading to relapses and recurrences. There is a risk of overstating the role of psychosis and delusional disorders provoking TESD. Despite the limited existence of clinical trials and the need for stronger evidence, from a clinical point of view, psychiatrists need a useful and practical approach to obtain appropriate coping strategies. The only way to know the true effect of medication is to properly explore sexual function both before and after starting a treatment with an antipsychotic. At the same time, knowing how relevant sexual aspects are for the patient’s quality of life could help us to sooner identify the risk of treatment-dropout due to poor tolerance of SD. Clinicians should monitor the possible changes in sexual function and should select an individualized strategy for the management of TESD that is poorly tolerated by the patient or their partner. Additionally, a rich and healthy sex life can help patients to maintain their bonds and contribute to much-needed socialization skills in psychosis.

Clinicians should be mainly interested in primary prevention and starting a treatment with APS that preserve sexual function should always be the first choice in sexually active patients who want to maintain their previous sexual activity. There is recent evidence that only 3% of psychiatrists routinely evaluate sexual functioning using specific psychometric tests [46], so psychiatrists’ interest in their patients’ sex life, as well as their specific skills to prevent, detect, and treat TESD, should be encouraged.

Aripiprazole, quetiapine, and ziprasidone, and perhaps olanzapine below 15 mg/day, are some of the better APS to either avoid or alleviate TESD. Risperidone, paliperidone, amisulpride, and typical APS should not be a first choice in patients with an active sex life who do not accept its deterioration. However, in some situations where it is not possible to avoid them, associating aripiprazole seems to be the best option. If prevention is not possible, there are several strategies to manage TESD symptomatology. The literature data reviewed in the present study (albeit scarce) [193], as well as recommendations based on the level of scientific evidence, allow us to recommend different intervention strategies according to the key TESD symptoms following the Scottish Intercollegiate Guidelines Network Grading Review Group [194] (Table 4).

## 5. Conclusions

The sexual function of psychotic patients must be assessed by the clinician thoroughly and regularly, in order to assess their tolerance to treatment with antipsychotics and avoid possible treatment dropouts. Some well validated scales for doing so are the PRSexDQ, ASFQ, and CSFQ, as they explore all stages of sexual functioning in patients with psychosis. Considering the current results, it seems clear that a subgroup of APS has a greater capacity for producing SD than others. Waiting for spontaneous TESD remission or reducing the antipsychotic dose are suboptimal strategies with little or no evidence and potential risk for psychotic relapse.

Aripiprazole (and perhaps other dopamine partial agonists like brexpiprazole or cariprazine) is the best option to prevent TESD, and it is the first-choice antipsychotic for this issue, in both antipsychotic-naïve patients and in those who have experienced SD as a consequence of treatment with another APS. Moreover, aripiprazole improves TESD and prolactin levels when it is used as coadjuvant treatment with another antipsychotic. Quetiapine or olanzapine (below 15 mg/day) can be useful second options for TESD in monotherapy when not surpassing a certain threshold dose. When switching or adding aripiprazole is not a viable option, PDE-5 inhibitors such as sildenafil, tadalafil, or vardenafil can improve ED in male patients.

Lastly, we should not forget that open communication with our patients about this problem is vital and, therefore, each management strategy needs to be individualized to the needs of each patient. Only if we do this will we be able to humanize treatment and improve patient–clinician relationships while ensuring adherence, quality of life, and the ability to maintain stable and enriching emotional bonds.

## 6. Limitations

There is little information compared with antidepressants on the best method of treatment for clinical evidence based TESD [148]. Some APS that are very similar to aripriprazole, like cariprazine or brexpiprazole, do not yet have studies on the subject and recommending them might be speculative. On the other hand, the disease itself may have some influence on the onset of sexual dysfunction and the antipsychotic may not be the only mechanism involved, as well as hyperprolactinemia. More research is needed in this field to obtain better recommendations based on clinical evidence.

## Figures and Tables

**Figure 1 jcm-10-00308-f001:**
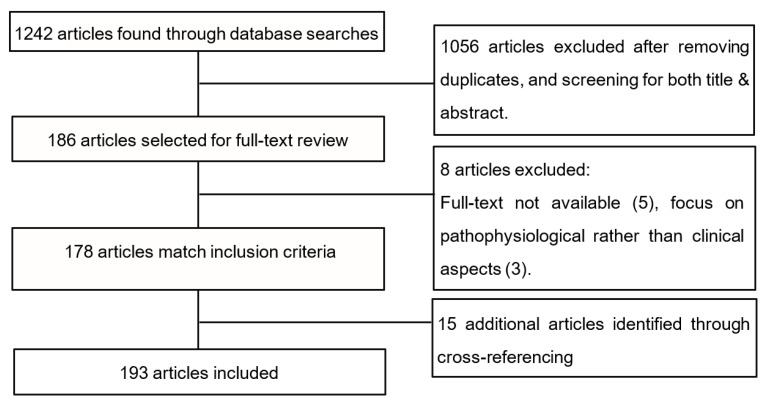
PRISMA flow diagram.

**Table 1 jcm-10-00308-t001:** Meta-analysis of the prevalence of sexual dysfunction (SD) in patients taking antipsychotics (APS) [61].

Antipsychotic	Prevalence of SD	Main Form of SD
Quetiapine (*n* = 1446)	16%	Desire (18%), arousal (12%), orgasm (7%)
Ziprasidone (*n* = 260)	18%	Desire (15%), arousal (18%), orgasm (19%)
Perphenazine (*n* = 261)	25%	Insufficient data
Aripiprazole (*n* = 62)	27%	Desire (12%), arousal (6%), orgasm (5%)
Olanzapine (*n* = 3521)	40%	Desire (24%), arousal (15%), orgasm (21%)
Risperidone (*n* = 1902)	43%	Desire (25%), arousal (21%), orgasm (22%)
Haloperidol (*n* = 364)	45%	Desire (27%), arousal (23%), orgasm (14%)
Clozapine (*n* = 110)	52%	Desire (37%), arousal (17%), orgasm (18%)
Thioridazine (*n* = 49)	60%	Arousal (46%), orgasm (49%)

**Table 2 jcm-10-00308-t002:** Mechanisms of action causing SD for each antipsychotic.

Antipsychotic	Mechanism of Action Causing SD	Endocrinological Changes	Dose at Which SD May Be More Likely to Appear
Haloperidol and other typical antipsychotics	Strong D2 antagonism and decrease of dopamine function.	Hyperprolactinemia and related changes (low testosterone, decreased estrogen levels, LH, and FSH)	Any
Risperidone, paliperidone	Strong D2 antagonism and decrease of dopamine function; weak 5HT2C antagonism	Hyperprolactinemia and related changes	Any
Amisulpride	Strong D2 antagonism and decrease of dopamine function	Hyperprolactinemia and related changes	Any
Clozapine	Weak D2 antagonism, mild 5HT2A antagonism, H1 and M1 antagonism	Mild hyperprolactinemia, indirect effects (sedation)	Variable
Ziprasidone	Strong 5HT2A antagonism and moderate D2 antagonism and 5HT2C antagonism	Transient hyperprolactinemia	Unknown
Olanzapine	Strong 5HT2A antagonism and moderate D2 and 5HT2C antagonism	Hyperprolactinemia at higher doses	> 15 mg/day
Quetiapine	Weak D2 antagonism, mild 5HT2A and 5HT2C antagonism	Hyperprolactinemia at higher doses	> 500 mg/day

LH, luteinizing hormone; FSH, follicle-stimulating hormone.

**Table 3 jcm-10-00308-t003:** Dosage correlation of treatment emergent sexual dysfunction (TESD) in Olanzapine and Risperidone at varying dosages [155].

Treatment	Dosage	*n*	% Sexual Dysfunction
Olanzapine *n* = 68	<15 mg	47	10.64
	>16 mg	21	33.33
Risperidone *n* = 128	<3 mg	35	48.6
	3–6 mg	48	81.3
	6–9 mg	16	75.0
	>9 mg	29	82.3

**Table 4 jcm-10-00308-t004:** Clinical recommendations for alleviating TESD based on scientific evidence levels * [194].

Symptom	Alternative 1	Evidence Level	Alternative 2	Evidence Level
Low sexual desire	Switching to aripiprazole	A	Adding aripiprazole	B
			Switching to olanzapine below 15 mg/day	B
	Switching to non hyperprolactinemic APS (quetiapine, ziprasidone)	B	Lowering the dose	C
Orgasm retardation	Switching to aripiprazole	A	Adding aripiprazole	B
			Switching to olanzapine below 15 mg/day	B
	Switching to non hyperprolactinemic APS (quetiapine, ziprasidone)	B	Lowering the dose	C
Anorgasmia	Switching to aripiprazole	A	Adding aripiprazole	B
			Switching to olanzapine below 15 mg/day	B
	Switching to non hyperprolactinemic APS (quetiapine, ziprasidone)	B	Lowering the dose	C
Erectile dysfunction	Switching to aripiprazole	A	Adding PD5 inhibitors	B
			Switching to olanzapine below 15 mg/day	B
	Switching to non hyperprolactinemic APS (quetiapine, ziprasidone)	B	Lowering the dose	C
Scarce vaginal lubrication	Switching to aripiprazole	A	Using vaginal lubricants	B
			Switching to olanzapine below 15 mg/day	B
	Switching to non hyperprolactinemic APS (quetiapine, ziprasidone)	B	Lowering the dose	C

* A: Recommended (good evidence that the measure is effective, and the benefits far outweigh the harms). B: Recommended (at least moderate evidence that the measure is effective, and the benefits outweigh the harms). C: Neither recommended, nor inadvisable (at least moderate evidence that the measure is effective; however, the level of benefit is very similar to the level of harm and a general recommendation cannot be justified).

## Data Availability

Not applicable.

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
