# Peer review of "Management Strategies for Antipsychotic-Related Sexual Dysfunction: A Clinical Approach"

_jcm, 2021, doi:10.3390/jcm10020308_

Round 1
Reviewer 1 Report
Review
Manuscript ID jcm-1004642
Title
Management Strategies for Antipsychotic-Related Sexual Dysfunction: A Clinical Approach
This is a well written article on a clinically highly relevant issue, the detection and management of antipsychotic related sexual dysfunction.
The authors have highlighted important areas and have structured the article suiting very well clinicians who wish to treat their patients while having in mind the side effect profiles of the antipsychotic medication.
I just have some critical remarks:
- Introduction:
The assessment of sexual dysfunctions (SD) is relevant for patients, not only regarding adherence, but also quality of life. The authors state, and I agree with that, that most often SD are underrecognized and they quote some studies on their prevalence or incidence. I would recommend, as SD develop in a very complex way comprising illness, comorbidity, addiction, medication, social, relationship and so on factors, that this should be mentioned. One of the pitfalls of all studies on SD is,that they do not disentangle this factors and their interplay with one another and just reduce the explanation to medication related side effect. And most often, assessment in the studies is very gross and does not include the symptom of suffering from these SD. This should be mentioned.
- Results
It should be mentioned that many patients do not have a monotherapy and this can also effect the emergence of SD.
- Intervention
Maybe waiting for improvement is associated with improvement of the mental disorder and as a consequence, the SD improves. This could also be an explanation why those with dose reduction improve. Maybe it is not only the dose itself, but the illness being better and allowing to reduce the dose. It should be clearly stated, that maybe in antidepressant treatment drug holiday, with all the risks of getting worse, could be recommended, but not in Schizophrenia spectrum disorders. Furthermore, the effect of drug holidays depends on the half-life of the used substances.
A table with the receptor profiles and a comment on how these receptors could cause SD and why these are to avoid or to use would be helpful.
Are there any data on the use of long-acting injections?
Generally, there are some typos and the English should be improved in the article.
Author Response
REVIEWER 1:
Dear reviewer.
We thank you very much for the time spent reviewing this article and the kind suggestions for improvement. Below you will find the answers to your comments and suggestions point by point.
- Introduction:
The assessment of sexual dysfunctions (SD) is relevant for patients, not only regarding adherence, but also quality of life. The authors state, and I agree with that, that most often SD are underrecognized and they quote some studies on their prevalence or incidence. I would recommend, as SD develop in a very complex way comprising illness, comorbidity, addiction, medication, social, relationship and so on factors, that this should be mentioned. One of the pitfalls of all studies on SD is,that they do not disentangle this factors and their interplay with one another and just reduce the explanation to medication related side effect. And most often, assessment in the studies is very gross and does not include the symptom of suffering from these SD. This should be mentioned.
This sentence “Although a recent study suggests that the severity of psychosis alone might contribute to SD independently of APS [49], most of the clinical literature is focused on this effect being a consequence of treatment rather than illness progression [50]” has been expanded into a full paragraph, that reads as follows: (line 76-82)
We must be mindful that SD usually develops in complex ways that involve several factors (comorbidity, culture, addictions, medication…) [42] and is the consequence of the interplay with one another. For example, a recent study suggests that the severity of psychosis alone might contribute to SD independently of APS [49]. The diversity of these factors can be often ignored or dismissed on SD clinical studies. However, most of the current clinical literature focuses on SD in psychotic patients because of suboptimal drug treatment [50], researching strategies to diminish or remove TESD from this perspective.
- Results
It should be mentioned that many patients do not have a monotherapy and this can also effect the emergence of SD.
We have included the following sentence: (line 194-195) “While many patients may not be on monotherapy and SD may emerge from a combination of drugs, prevention allows us to optimize the risk for SD when treating an antipsychotic-naïve patient.”
- Intervention
Maybe waiting for improvement is associated with improvement of the mental disorder and as a consequence, the SD improves. This could also be an explanation why those with dose reduction improve. Maybe it is not only the dose itself, but the illness being better and allowing to reduce the dose. It should be clearly stated, that maybe in antidepressant treatment drug holiday, with all the risks of getting worse, could be recommended, but not in Schizophrenia spectrum disorders. Furthermore, the effect of drug holidays depends on the half-life of the used substances.
The following sentences have been added:
- Section 3.3.1: “An alternative explanation for SD spontaneously improving over time would be that SD is more directly related to the severity of illness rather than the antipsychotic treatment; and thus, after being treated, sexual function would improve consequently.”
- Section 3.3.2: “An alternative explanation could be that illness severity is related to SD, and since the decision to lower dosage implies a clinical improvement, psychosis may not impact sexual function as severely as severely as previous higher doses of antipsychotic.”
The last part of section 3.3.2 has been rewritten as follows: “A strategy similar to dose reduction that has sometimes worked when dealing with TESD caused by antidepressants is the periodic treatment interruption or “weekend holiday” [148] where the patient interrupts treatment for a period of time (24 hours prior to intercourse). The usefulness of this strategy is highly dependent on the specific half-life of the drug being used. No studies have been performed with APS regarding this strategy [2,156]. In fact, randomized controlled trials are currently needed to provide evidence for this or any other dose reduction strategies [147,157].
While perhaps useful regarding antidepressant treatment, this strategy is not recommended in Schizophrenia spectrum disorders as it carries significant risks of relapse and may inadvertently encourage treatment drop-out.”
A table with the receptor profiles and a comment on how these receptors could cause SD and why these are to avoid or to use would be helpful.
A new table detailing each antipsychotic mechanism of causing SD has been added at the end of Section 3.1.
Are there any data on the use of long-acting injections?
The sentence in Section 3.3.3 that read “This has been observed both in oral and long-acting presentations [106]” has been rewritten as follows: “The only study assessing the issue of TESD in long-acting intramuscular presentations [106] corroborates this recommendation when comparing aripiprazole to paliperidone palmitate.”
Generally, there are some typos and the English should be improved in the article.
A thorough collective re-read of the paper has been performed in order to correct typos, spelling and grammar mistakes.
Reviewer 2 Report
In this article Montejo and colleagues give clear recommendations for the management of antipsychotic-related sexual dysfunction. They have thoroughly explored all available data and made the effort to include even small case studies. Multiple resources were used to support given facts. Whereas section 3.1 is a bit hard to read and lacks structure, the other sections of this article are easily interpreted and give a clear overview. Clear recommendations are being provided, making this review engaging to be used in clinical practice. However, these recommendations do not all logically result from previous sections of the paper.
Some recommendations for improvement
General
- The authors do not mention it explicitly, but they have performed a qualitative literature review including a database search. Including a PRISMA schema would be very informative and also make the paper methodologically more sound by following those rules.
- Overall, quite some typo’s, spelling and grammar mistakes were found in this paper. A simple, careful, reread could effectively improve the quality of this paper.
- Throughout the whole article, capital versus small letters are inconsistently used for generic drugs.
- More examples of inconsistencies can be found throughout the text; for example, treatment emergent sexual dysfunction in the introduction and treatment-emergent sexual dysfunction in the methods section.
Abstract
- The following phrase from lines 21-22 is suboptimal: ‘prescribing either aripiprazole or a partial dopamine partial agonist’. It suggests that aripiprazole is not a partial dopamine agonist (which it is indeed) and the word ‘partial’ is being used twice.
Introduction
- Maybe the sentence in lines 82-84 could be rewritten, since its reasoning is illogical.
Methods
- The overview of databases used to retrieve articles is incomplete; results would be better reproducible if all used databased were stated.
- Search strings were specific for erectile dysfunction and anorgasmia, but not for other sexual dysfunctions such as ‘desire’, even though this is, according to table 1, the most frequently reported sexual dysfunction.
Results
- Maybe lines 167-171 could be rewritten, since it is stated twice that quetiapine is compared to olanzapine.
- 1: this section requires quite careful reading in order to get a clear overview. Perhaps these results could be provided more schematically, for example by the use of a table.
Clinical recommendations
- The recommendations in table 4 do not all follow the results section. For example, a recommendation is being made to switch to aripiprazole in case of low sexual desire, however, this has not been explicitly stated elsewhere. It would be more logical if all recommendations would clearly follow the results section.
Conclusions
- It is stated that no information about the effects of PDE-5 inhibitors on sexual dysfunctions other than erectile dysfunction is available; this implies that PDE-5 inhibitors could have effects on multiple kinds of sexual dysfunction. However, considering the mechanism of action of these drugs, this also does not seem logical.
Author Response
REVIEWER 2:
In this article Montejo and colleagues give clear recommendations for the management of antipsychotic-related sexual dysfunction. They have thoroughly explored all available data and made the effort to include even small case studies. Multiple resources were used to support given facts. Whereas section 3.1 is a bit hard to read and lacks structure, the other sections of this article are easily interpreted and give a clear overview. Clear recommendations are being provided, making this review engaging to be used in clinical practice. However, these recommendations do not all logically result from previous sections of the paper.
Dear reviewer.
We thank you very much for the time spent reviewing this article and the kind suggestions for improvement. Below you will find the answers to your comments and suggestions point by point.
Some recommendations for improvement
General
- The authors do not mention it explicitly, but they have performed a qualitative literature review including a database search. Including a PRISMA schema would be very informative and also make the paper methodologically more sound by following those rules.
A PRISMA flow diagram has been included in the methodology section (line 167).
- Overall, quite some typo’s, spelling and grammar mistakes were found in this paper. A simple, careful, reread could effectively improve the quality of this paper.
A thorough collective re-read of the paper has been performed in order to correct typos, spelling and grammar mistakes.
- Throughout the whole article, capital versus small letters are inconsistently used for generic drugs.
Generic drugs spelling is now homogenized with small letter spelling throughout the paper.
- More examples of inconsistencies can be found throughout the text; for example, treatment emergent sexual dysfunction in the introduction and treatment-emergent sexual dysfunction in the methods section.
A thorough collective re-read of the paper has been performed in order to correct inconsistencies. The term “treatment-emergent sexual dysfunction” in the methods section has been replaced for “treatment emergent sexual dysfunction” (without hyphen).
Abstract
- The following phrase from lines 21-22 is suboptimal: ‘prescribing either aripiprazole or a partial dopamine partial agonist’. It suggests that aripiprazole is not a partial dopamine agonist (which it is indeed) and the word ‘partial’ is being used twice.
This sentence has been rewritten as follows: “prescribing either aripiprazole or another partial dopamine agonist”.
Introduction
- Maybe the sentence in lines 82-84 could be rewritten, since its reasoning is illogical.
The original sentence has been rewritten as follows: “All these reasons contribute to the incidence of SD being underestimated and underreported [46,56,57]. This issue going unnoticed can cause difficulties in interpersonal relationships and sexual dissatisfaction for patients and their partners”.
Methods
- The overview of databases used to retrieve articles is incomplete; results would be better reproducible if all used databased were stated.
In the sentence “PubMed and Cochrane were the main databases used”, the word “main” has now been removed. Our initial goal was to include Google scholar as well, but time constraints kept this from happening.
- Search strings were specific for erectile dysfunction and anorgasmia, but not for other sexual dysfunctions such as ‘desire’, even though this is, according to table 1, the most frequently reported sexual dysfunction.
We have added “desire” as a specific search term. An oversight on our part.
Results
- Maybe lines 167-171 could be rewritten, since it is stated twice that quetiapine is compared to olanzapine.
The lines have been rewritten as follows: “Quetiapine does not significantly increase prolactin levels [82]. There has been fewer reported incidence of SD with Quetiapine when compared to typical neuroleptics, and some atypical (risperidone and olanzapine) [72,83,84,119].”
- 1: this section requires quite careful reading in order to get a clear overview. Perhaps these results could be provided more schematically, for example by the use of a table.
Section 3.1 has been slightly rewritten in order to give a clear overview of each drug and their relation to SD.
A new table detailing each antipsychotic mechanism of causing SD has been added at the end ( Table 2).
Clinical recommendations
- The recommendations in table 4 do not all follow the results section. For example, a recommendation is being made to switch to aripiprazole in case of low sexual desire, however, this has not been explicitly stated elsewhere. It would be more logical if all recommendations would clearly follow the results section.
Part of Section 3.3.4 has been rewritten as follows “Most authors agree on aripiprazole as the best option in this case [1,89,119,134,150,157] since several studies have proven that improves SD rates across all sexual dimensions (desire, arousal and orgasm) and decreases prolactin […]”
Conclusions
- It is stated that no information about the effects of PDE-5 inhibitors on sexual dysfunctions other than erectile dysfunction is available; this implies that PDE-5 inhibitors could have effects on multiple kinds of sexual dysfunction. However, considering the mechanism of action of these drugs, this also does not seem logical.
The line in Section 5 (Conclusions) concerning PDE-5 inhibitors that read “However, there is not information about its effects in the improvement of the other areas of sexual life such as sexual interest or orgasm quality” has been removed.

Round 2
Reviewer 2 Report
First, I would like to express my appreciation for the improvements that the authors have made regarding their article. Overall, its quality has been notably improved. The article is easier to read and is of higher value content wise. The addition of table 2 gives a great overview of section 3. Following are a few minor suggestions for further improvement: - Lines 72-76, 77-78 and 205 could be grammatically improved. - The revision of lines 286-289 is fine content wise , but needs a spell check